# Observational descriptive study of ultrasound use and its impact on clinical decisions in the accident and emergency department at Georgetown public hospital corporation

**Davendra Vishaul Kissoon**[1], **Sri Devi Jagjit**[1], **Brian D. Bales**[2], **Zelda Luke-Blyden**[1], **Jeremy S. Boyd**[2], **Jordan D. Rupp**[2]*

**1** Accident & Emergency Department, Georgetown Public Hospital Corporation, Georgetown, Guyana, **2** Department of Emergency Medicine, Vanderbilt University Medical Center, Nashville, Tennessee, United States of America

* jordan.rupp@vumc.org

## Abstract

### Introduction

Point-of-care ultrasound (POCUS) is physician-performed at the bedside, and it is a powerful diagnostic tool, especially in resource-limited emergency medicine healthcare settings. This study aims to quantify both the use of ultrasound and its impact on patient care at the Accident and Emergency Department at the Georgetown Public Hospital Corporation (GPHC).

### Methods

This is a cross-sectional observational descriptive analysis of data collected for quality assessment in the GPHC Accident and Emergency Department. Over the course of two months, physicians were asked to record each ultrasound exam performed and record whether the ultrasound results changed patient disposition or the medication used in management.

### Results

During the study period, there were 173 ultrasound data sheets collected representing 426 ultrasound studies. 196 studies were positive with pathologic findings (46.0%). The use of ultrasound in patient care either changed the patient's final disposition or medication 78.6% of the time.

### Conclusion

Ultrasound is used frequently at the Georgetown Public Hospital Corporation for a wide variety of applications. When utilized, POCUS frequently influenced patient care.

**Data Availability Statement:** The data has been shared in a public depository. It can be found at the

following location: dx.doi.org/10.17504/protocols.io.bd6yi9fw

**Funding:** The authors received no specific funding for this research study.

**Competing interests:** The authors have declared that no competing interests exist.

## Introduction

Ultrasound is an important diagnostic tool in medicine. It has a wide accessibility because it is portable, versatile, and easy to use. Point-of-care ultrasound (POCUS) has been developed because of these qualities. [1] POCUS is performed by the physician immediately at the bedside. POCUS was first developed in the emergency department due to the presence of undifferentiated and critically ill patients, the need for rapid diagnosis, and the ability of the physician to readily utilize ultrasound once proper training is attained. [2]

POCUS helps to improve the overall outcomes of patient care in emergency departments. [3] A retrospective study done in Tanzania showed that clinicians changed their diagnostic impression and plan in 29% of cases after an ultrasound was performed, and disposition plan was changed in 45% of the patients after an ultrasound was performed. Patient care was greatly impacted by the use of ultrasound in the emergency setting in Tanzania. [4] Point-of-care ultrasound's use, accuracy and impact on clinical decision making were measured in a hospital in Rwanda where there is a continuous training program in place. Ultrasound changed medical decision-making 81.3% of the time, most frequently medication administration (42.4%) and admission (30%). [5] There is additional data from a smaller study in Liberia. It showed that ultrasound changed patient management in 62% of cases. These studies were performed throughout the hospital. Only 28% of these cases were in the emergency department. [6]

Georgetown Public Hospital Corporation (GPHC) is the tertiary referral hospital in the public health system in Guyana, a middle-income country in South America. The public system is resource-limited, and there is limited access to advanced imaging. The Accident and Emergency Department evaluated 40,000 patients in 2019. GPHC does an excellent job of identifying low acuity cases that can be evaluated in their medicine clinics. The department cares for both adult and pediatric patients. It also cares for obstetric patients until viability. GPHC has had an emergency medicine residency-training program in partnership with the University of Guyana since 2010. Shortly after the start of the program, POCUS was introduced to the GPHC Accident & Emergency Department. Beginning in 2015, a formal ultrasound-training curriculum was introduced. [7] Residency trainees are formally trained and credentialed to use ultrasound in patient care.

POCUS has substantial impact in the GPHC Accident & Emergency Department. With the introduction of POCUS and proper training of personnel, clinical decisions are made more rapidly leading to more accurate and timely treatments. The impact of physician-performed ultrasound has never been documented or quantified. This study aims to quantify both the use of ultrasound and its impact on patient care at the Accident and Emergency Department at the Georgetown Public Hospital Corporation.

## Methods

This is a cross-sectional observational descriptive analysis of a departmental ultrasound usage in the GPHC Accident and Emergency Department. The study period began with the start of night shift on 31 May 2019 until the end of day shift on 31 July 2019. Physicians were asked to record each ultrasound exam performed during their shift. With each ultrasound exam, the physicians were asked to record the type of exam, the result of the exam (positive or negative), impact on disposition (yes or no), and impact on medication administered (yes or no). Intravenous fluids were considered a medication for the purposes of this study. If the disposition of the patient was changed due to the ultrasound findings, physicians were also asked to record the final disposition plan (theatre, admission, discharge, etc.).

The information from the data collection forms was entered into Excel (Microsoft–Redmond, Washington, USA) along with information about the providers' level of training. There

are three types of participating physician-providers in the Accident & Emergency Department: registrars (completed emergency medicine specific residency training), residents (currently training in emergency medicine), and general medical officers (no specific emergency medicine training). Information about the number of patients seen per provider was also collected. Further departmental statistics were used to help understand the data quality and completeness: patient volume during each month and number of physician specific shifts during each month. Data is reported either directly or as simple proportions and percentages.

This study describes the use of ultrasound by physicians in the GPHC Accident & Emergency Department. GPHC is a large, tertiary health care center. It serves as a central referral hospital for the nation's public hospital system. All physicians were asked to participate. Physicians chose which patients to ultrasound based on their standard practice patterns. There were no specific exclusion criteria. As data was collected as a part of departmental descriptive analysis to help inform future budget proposals and future training, patient identifiers were not collected, and consent was waived. The GPHC institutional review board has approved publication of the data (IRB FWA00014641 protocol #586/2019). This study was written in accordance with the Strengthening the Reporting of Observational Studies in Epidemiology Guidelines. [8]

## Results

During the two-month study period, there were a total of 173 ultrasound data sheets collected. 426 ultrasound studies were performed during these 173 shifts with an average of 2.46 ultrasounds per shift (See Table 1). The most commonly performed ultrasound exams were EFAST (115), Obstetric (100), and Cardiac (91).

Physicians averaged 0.283 ultrasound exams per hour of work. Physicians did not consistently record the number of patients seen. During the 107 shifts were they were recorded, 282 ultrasounds were performed. This means 0.287 ultrasound exams were performed per patient.

Of the 426 ultrasound studies performed, 196 had pathologic findings (46.0%). The use of ultrasound changed the final patient disposition for 276 patients (64.8%). Specifically, the use of ultrasound aided the decision to admit (22.8%), to discharge (22.8%), to involve specialist review (14.3%), and to go to operating theatre (23.4%). Physicians recorded a change in medication administration based on ultrasound findings in 58.9%. The use of ultrasound in patient care either effected the patient's final disposition or medication 78.6% of the time (See Table 2).

Notable subset analysis showed that 26 EFAST exams identified positive findings (23.0%). For obstetric patient care, the ultrasound was the pivotal piece of information determining patient disposition in 88.0% of cases. This was true whether the ultrasound had pathologic findings or showed a viable intrauterine pregnancy. Lung ultrasound appears to be under-utilized. It was positive every time it was used (15/15).

**Table 1. Overall data.**

|  | Total | June | July |
|---|---|---|---|
| Ultrasound datasheets | 173 | 126 | 38* |
| Ultrasounds performed | 426 | 302 | 111 |
| Average ultrasound per shift | 2.46 | 2.40 | 2.92 |
| A&E patient volume | 7711 | 3816 | 3895 |
| Ultrasound per patient seen | 282/982 (28.7%) |  |  |

*9 data sheets were not dated properly

**Table 2. Overall effect of ultrasound on patient care.**

| | Number of Studies | Positive | Change in Disposition | Admit | Discharge | Specialist Review | Theatre | Change in Medication | Change in Care |
|---|---|---|---|---|---|---|---|---|---|
| Total | 426 | 196 (46.0%) | 276 (64.7%) | 97 (22.8%) | 97 (22.8%) | 61 (14.3%) | 10 (2.35%) | 251 (58.9%) | 335 (78.6%) |
| Abdomen | 45 | 12 (26.7%) | 28 (62.2%) | 7 | 12 | 3 | 1' | 27 (60.0%) | 32 (71.1%) |
| Aorta | 3 | 0 | 1 (33.3%) | 0 | 1 | 0 | 0 | 1 (33.3%) | 2 (66.6%) |
| Biliary | 14 | 6 (42.9%) | 6 (42.9%) | 3 | 2 | 1 | 0 | 7 (46.6%) | 9 (64.3%) |
| Cardiac | 93 | 56 (60.2%) | 59 (63.4%) | 31 | 9 | 15 | 0 | 60 (63.8%) | 75 (80.6%) |
| DVT | 5 | 4 (80.0%) | 3 (60.0%) | 2 | 0 | 1 | 0 | 3 (60.0%) | 4 (80.0%) |
| EFAST | 113 | 26 (23.0%) | 56 (49.6%) | 20 | 22 | 7 | 7 | 59 (52.2%) | 76 (67.3%) |
| Lung | 15 | 15 (100%) | 12 (80.0%) | 6 | 2 | 4 | 0 | 12 (80.0%) | 15 (100%) |
| Obstetric | 100 | 63 (63.0%) | 88 (88.0%) | 22 | 38 | 25 | 2 | 64 (64.0%) | 93 (93.0%) |
| Pelvic | 22 | 5 (22.7%) | 13 (59.1%) | 4 | 6 | 3 | 0 | 9 (40.9%) | 17 (77.3%) |
| Renal | 3 | 3 (100%) | 2 (66.7%) | 1 | 0 | 0 | 0 | 2 (66.7%) | 3 (100%) |
| Soft Tissue | 4 | 3 (75.0%) | 3 (75.0%) | 0 | 3 | 0 | 0 | 3 (75.0%) | 3 (75.0%) |
| Vascular | 3 | 2 (66.7%) | 2 (66.7%) | 0 | 1 | 1 | 0 | 1 (33.3%) | 2 (66.7%) |
| Other | 4 | 0 | 2 (50.0% | 1 | 1 | 0 | 0 | 2 (50.0%) | 4 (100%) |

A significant challenge was participation of all the physicians filling out forms every shift throughout the two months. In total, there were 1024 shifts worked over the study period of which there were 173 shift sheets gathered (16.9%). Providers did not always record their level of training. GMOs worked 452 shifts and recorded data on 52 shift sheets (11.5%). Residents recorded data sheets on 79 of 251 shifts worked (31.5%), and registrars recorded data sheets on 13 of the 321 shifts worked (4.05%). The limited number of ultrasounds recorded by registrars was expected. The registrars function in a supervisory role for the residents and GMOs including supervising many of the ultrasounds performed. The registrars may not see any patients primarily and were advised to only record ultrasounds on their own patients. Details are included in Table 3.

## Discussion

This data shows POCUS frequently determined patient care decisions in the Georgetown Public Hospital Accident and Emergency Department. Ultrasound is utilized commonly to determine diagnoses or rule out emergent diagnoses. Most importantly, POCUS helps to manage patient care by frequently aiding in the disposition and aiding in the medications administered to the patient. Quantifying the benefits of ultrasound will help identify gaps in training, and help to identify the utility of investing further in ultrasound equipment in the future.

This data reveals that ultrasound is most frequently used in the care of trauma patients, obstetric patients, and patients with possible cardiac related symptoms. Despite limitations of reporting, there was a positive EFAST exam 26 times in 61 days of data collection, 7 of which

**Table 3. Ultrasound frequency by level of training.**

| | Ultrasound Total | Ultrasound per Shift | Total Hours worked | Ultrasound per Hour |
|---|---|---|---|---|
| Total | 411 | 2.46 | 1452 | 0.283 |
| GMO | 126 | 2.25 | 452 | 0.279 |
| Resident | 235 | 2.61 | 772 | 0.304 |
| Registrar | 50 | 2.63 | 228 | 0.219 |

went directly to the operating theatre. The physician-performed ultrasound is the key piece of information in the care of pregnant patients presenting to the GPHC Accident & Emergency Department. 88.0% of the time the ultrasound changed the pregnant patient's disposition, sometimes confirming a safe discharge and other times prompting admission or surgery. Based on this data, lung ultrasound is under-utilized in the department at this time. It was only utilized 15 times during the study period and was positive every time. This suggests the lung ultrasound should be utilized more frequently to guide patient management.

The impact of ultrasound on patient care at the GPHC Accident & Emergency Department was consistent with a previous study performed in Africa. Ultrasound altered patient treatment 78.6% of the time at GPHC compared to 81.3% in the previously cited study from Rwanda.[5] The data from this study would be most applicable to other central public referral hospitals with similar resources.

The biggest limitation to this study is the sample size. Only 16.9% of the shifts were recorded. The registrars were the least likely to report data. With the supervising role, the registrars evaluate a very small portion of the patients primarily. The smaller sample size creates more room for bias, particularly reporting bias. Physicians that frequently use ultrasound may be more likely to report their usage. A data point that was not consistently recorded by the physicians was the total number of patients seen during each shift; therefore, our data on ultrasound studies per patient seen is further limited. Finally, there is individual subjectivity by each participating physician in regards to whether the ultrasound changed their medical decision-making.

## Conclusion

Ultrasound is commonly utilized at the Georgetown Public Hospital Corporation for a wide variety of applications. When conducted, POCUS frequently influenced patient care.

## Supporting information

**S1 File. Ultrasound usage: GPHC.**
(PDF)

## Author Contributions

**Conceptualization:** Davendra Vishaul Kissoon, Sri Devi Jagjit, Brian D. Bales, Zelda Luke-Blyden, Jeremy S. Boyd, Jordan D. Rupp.

**Data curation:** Davendra Vishaul Kissoon, Sri Devi Jagjit, Jordan D. Rupp.

**Formal analysis:** Davendra Vishaul Kissoon, Sri Devi Jagjit, Brian D. Bales, Jeremy S. Boyd, Jordan D. Rupp.

**Investigation:** Davendra Vishaul Kissoon, Sri Devi Jagjit.

**Methodology:** Davendra Vishaul Kissoon, Sri Devi Jagjit, Zelda Luke-Blyden, Jordan D. Rupp.

**Project administration:** Davendra Vishaul Kissoon, Zelda Luke-Blyden, Jordan D. Rupp.

**Supervision:** Zelda Luke-Blyden, Jordan D. Rupp.

**Writing – original draft:** Davendra Vishaul Kissoon, Jordan D. Rupp.

**Writing – review & editing:** Davendra Vishaul Kissoon, Sri Devi Jagjit, Brian D. Bales, Zelda Luke-Blyden, Jeremy S. Boyd.

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
