## [Decision Letter · Decision Letter 0]

9 Mar 2020

PONE-D-19-36065

Ultrasound Use and its Impact on Clinical Decisions in the Accident and Emergency Department at Georgetown Public Hospital Corporation

PLOS ONE

Dear Dr. Rupp,

Thank you for submitting your manuscript to PLOS ONE. After careful consideration, we feel that it has merit but does not fully meet PLOS ONE’s publication criteria as it currently stands. Therefore, we invite you to submit a revised version of the manuscript that addresses the points raised during the review process.

We would appreciate receiving your revised manuscript by Apr 23 2020 11:59PM. To enhance the reproducibility of your results, we recommend that if applicable you deposit your laboratory protocols in protocols.io, where a protocol can be assigned its own identifier (DOI) such that it can be cited independently in the future. For instructions see: http://journals.plos.org/plosone/s/submission-guidelines#loc-laboratory-protocols

We look forward to receiving your revised manuscript.

Kind regards,

Andrew Carl Miller

Academic Editor

PLOS ONE

Journal Requirements:

Reviewers' comments:

Reviewer's Responses to Questions

**Comments to the Author**

1. Is the manuscript technically sound, and do the data support the conclusions?

Reviewer #1: Yes

Reviewer #2: Yes

Reviewer #3: Yes

Reviewer #4: Yes

2. Has the statistical analysis been performed appropriately and rigorously? 

Reviewer #1: Yes

Reviewer #2: Yes

Reviewer #3: Yes

Reviewer #4: No

3. Have the authors made all data underlying the findings in their manuscript fully available?

Reviewer #1: Yes

Reviewer #2: Yes

Reviewer #3: Yes

Reviewer #4: Yes

4. Is the manuscript presented in an intelligible fashion and written in standard English?

Reviewer #1: Yes

Reviewer #2: Yes

Reviewer #3: Yes

Reviewer #4: Yes

5. Review Comments to the Author

Reviewer #1: Overall, a well done study and clear, concise write-up.

A few suggestions and comments:

A little more detail on the facility would give the reader a better understanding - are pediatric cases seen here? What is the annual volume?

Would love to see any further data on the obstetric cases - are they primarily 1st trimester problems or later pregnancy complications?

While it may not be possible in this case, a more detailed analysis of how POCUS changed management would be useful. Were the EFASTs primarily hypotensive patients with intra-peritoneal bleeding? Signs of cardiac injuries? For the cardiac cases, was it primarily findings of pericardial effusions? Cardiomyopathies?

Any data on POCUS guided procedures?

Reviewer #2: Descriptive study that attempts to answer a complicated question that is difficult to truly study. The study was done in an acceptable way. It is explained well. Data supports conclusions. Certainly not an earth-shattering study, but does make a useful contribution to the literature.

Reviewer #3: Thank you for this interesting submission. Your study is similar to a few other studies, previously published, which were not specifically referenced in your paper but had very similar results:

1. Shah SP, Epino H, et al. Impact of the introduction of ultrasound services in a limited resource setting: rural Rwanda 2008. BMC Int Health Hum Rights. 2009 Mar 27;9:4

and

2. Kotlyar S, Moore CL. Assessing the utility of ultrasound in Liberia. J Emerg Trauma Shock. 2008 Jan;1(1):10-4.

I do have a few questions regarding your submission.

First, is there any QI process for the images and decision making in this hospital? This may skew your results as trainees and novice users may misinterpret images and this may affect clinical decision making as well.

Second, was data collected on the use of POCUS altering the decision to obtain further imaging/diagnostic testing? This is a valuable end-point, and one which is addressed in prior studies.

Reviewer #4: - Please insert page numbers.

- Please insert line numbers.

- Indicate the study’s design with a commonly used term in the title.

- The authors state that this is a quality improvement study, however it is not. It is simply an observational descriptive analysis. In order to be a QI study, it must concern an initiative to improve healthcare (broadly defined to include the quality, safety, effectiveness, patient centeredness, timeliness, cost, efficiency, and equity of healthcare). It’s strictly describing the scope of ultrasound use in their ED.

- Please ensure that the manuscript adheres to the proper guideline. If an observational study, STROBE (PMID: 17941714).

- If truly a QI study, then SQUIRE guidelines (PMID: 26369893). Be sure to provide appropriate citation.

- Please include the IRB approval number in parentheses.

- Explain how the study size was arrived at.

- Statistical methods should be described.

- Is the level of US use similar or different to other academic programs in your country? What about to non-academic emergency departments?

6. PLOS authors have the option to publish the peer review history of their article (what does this mean?). If published, this will include your full peer review and any attached files.

Reviewer #1: No

Reviewer #2: No

Reviewer #3: No

Reviewer #4: No

---

## [Author Response · Author response to Decision Letter 0]

15 Apr 2020

15 April 2020

To Whom It May Concern:

Thank you for your reviews of the submitted article, Observational Descriptive Study of Ultrasound Use and its Impact on Clinical Decisions in the Accident and Emergency Department at Georgetown Public Hospital Corporation. Included in this letter are responses to concerns by the initial reviewers. The original article with edits is also included with this resubmission along with the article with the changes fully included.

The requested journal requirements have been addressed as follows:

1. Adjustments have been made to the formatting. There are no directions on file naming based in the link provided.

2. In response to the data, the data has been shared in a public depository. It can be found at the following location: dx.doi.org/10.17504/protocols.io.bd6yi9fw.

3. The caption for the supplementary file is included in the supporting information section.

In regards to the Review Comments to the Author, responses are addressed as follows:

Reviewer #1

-Additional detail regarding the described treatment facility has been included n the manuscript.

-Specific details about the gestational ages of the pregnancies is not available to report. The issues are primarily first trimester issues and issues related to failed pregnancies (retained products of conception).

-For the EFAST exam, the primary positive findings were pneumothorax, hemothorax, and intraperitoneal bleeding. For cardiac causes, decreased left ventricular function was the predominant finding. For each application, the positive findings were incompletely recorded and therefore was not reported. 

-POCUS guided procedures are very rare. Central lines kits are not always available. Thoracentesis and paracentesis are usually performed on the medicine wards where there is more physical space to perform the procedure safely.

Reviewer #2

-There are no questions to address from Reviewer #2.

Reviewer #3

-We reviewed both of the mentioned articles prior to our initial submission, Shah et al and Kotlyar and Moore. The article by Shah et al was done in rural hospitals and does not clearly state the clinical setting. Specifically, it is not in the same clinical setting as our study, the emergency department. The Kotlyar and Moore study involved all departments of the hospital. Only 28% of the patients were in the emergency department. It was not initially included for this reason, but we have added per the reviewer’s request.

- The goal of this study was to quantify the use of ultrasound in the A&E. There is an established training program for the residents. There is ongoing quality assessment of all scans performed by the GMOs and residents in real time with the registrars and visiting ultrasound-trained faculty. There are no current mechanisms for data storage and ongoing review of the registrars’ studies. Each registrar has completed the ultrasound certification.

- The availability of advanced imaging beyond ultrasound is severely limited and rarely used. The clinical decision for advanced imaging is usually made by the consulting service (ex. general surgery) because many patients cannot afford it and it takes a large amount of time.

Reviewer #4

-Line numbers have been inserted.

-The study design has been more clearly communicated in both the title and body of the manuscript.

-The description of the study has been updated to more accurately describe the scope of our study.

-This paper was clarified as an observational study. We have reviewed the STROBE guidelines and ensured that our manuscript fits within the recommendations. The citation has been included in the references.

-The SQUIRE guidelines are no longer applicable to this study.

-The IRB approval number has been included in the manuscript.

-The two month sample was felt to be an adequate cross section to reflect ultrasound use at GPHC. We used the similar cited studies that were performed at clinical sites in Africa to help guide our sample size.

-Statistical methods have been clarified. Simple proportions were used to report the data.

- Similar data from a U.S. training site was not available for direct comparison. In regards to non-academic sites in Guyana, there is currently no data available. GPHC is the only tertiary center. The remaining regional health facilities have very limited resource. One site just recent acquired an ultrasound device for the A&E, and training is on-going. 

Thank you,

Davendra Vishaul Kissoon, MD

Jordan Rupp, MD

Corresponding Author:

Jordan Rupp, MD, FACEP

1313 21st Avenue South

Oxford House 703

Nashville, TN 37212

(t) 615-936-0087

(m) 4190377-7526

(f) 615-936-1316

jordan.rupp@vumc.org

---

## [Decision Letter · Decision Letter 1]

5 May 2020

Observational Descriptive Study of Ultrasound Use and its Impact on Clinical Decisions in the Accident and Emergency Department at Georgetown Public Hospital Corporation

PONE-D-19-36065R1

Dear Dr. Rupp,

We are pleased to inform you that your manuscript has been judged scientifically suitable for publication and will be formally accepted for publication once it complies with all outstanding technical requirements.

With kind regards,

Andrew Carl Miller

Academic Editor

PLOS ONE

Additional Editor Comments (optional):

The reviewers have addressed all expressed concerns and queries. At this point I recommend acceptance.

Reviewers' comments:

Reviewer's Responses to Questions

**Comments to the Author**

1. If the authors have adequately addressed your comments raised in a previous round of review and you feel that this manuscript is now acceptable for publication, you may indicate that here to bypass the “Comments to the Author” section, enter your conflict of interest statement in the “Confidential to Editor” section, and submit your "Accept" recommendation.

Reviewer #1: All comments have been addressed

Reviewer #4: All comments have been addressed

2. Is the manuscript technically sound, and do the data support the conclusions?

Reviewer #1: Yes

Reviewer #4: Yes

3. Has the statistical analysis been performed appropriately and rigorously? 

Reviewer #1: Yes

Reviewer #4: Yes

4. Have the authors made all data underlying the findings in their manuscript fully available?

Reviewer #1: Yes

Reviewer #4: Yes

5. Is the manuscript presented in an intelligible fashion and written in standard English?

Reviewer #1: Yes

Reviewer #4: Yes

6. Review Comments to the Author

Reviewer #1: (No Response)

Reviewer #4: The reviewers have addressed all expressed concerns and queries. At this point I recommend acceptance.

7. PLOS authors have the option to publish the peer review history of their article (what does this mean?). If published, this will include your full peer review and any attached files.

Reviewer #1: No

Reviewer #4: Yes: Andrew C. Miller

---

## [Editor Report · Acceptance letter]

6 May 2020

PONE-D-19-36065R1 

Observational Descriptive Study of Ultrasound Use and its Impact on Clinical Decisions in the Accident and Emergency Department at Georgetown Public Hospital Corporation 

Dear Dr. Rupp:

I am pleased to inform you that your manuscript has been deemed suitable for publication in PLOS ONE. Congratulations! Your manuscript is now with our production department. 

With kind regards,

on behalf of

Dr. Andrew Carl Miller 

Academic Editor

PLOS ONE